# RELATIONAL CONSTRAINTS ON NEURAL NETWORKS REPRODUCE HUMAN BIASES TOWARDS ABSTRACT GEOMETRIC REGULARITY

## ABSTRACT

Uniquely among primates, humans possess a remarkable capacity to recognize and manipulate abstract structure in the service of task goals across a broad range of behaviors. One illustration of this is in the visual perception of geometric forms. Studies have shown a uniquely human bias toward geometric regularity, with task performance enhanced for more regular and symmetric forms compared to their geometrically irregular counterparts. Such studies conclude that this behavior implies the existence of discrete symbolic structure in human mental representations, and that replicating such behavior in neural network architectures will require mechanisms for symbolic processing. In this study, we argue that human biases towards geometric regularity can be reproduced in neural networks, without explicitly providing them with symbolic machinery, by augmenting them with an architectural constraint that enables the system to discover and manipulate relational structure. When trained with the appropriate curriculum, this model exhibits human-like biases towards symmetry and regularity in two distinct tasks involving abstract geometric reasoning. Our findings indicate that neural networks, when equipped with the necessary training objectives and architectural elements, can exhibit human-like regularity biases and generalization. This approach provides insights into the neural mechanisms underlying geometric reasoning and offers an alternative to prevailing symbolic "Language of Thought" models in this domain.

## 1 INTRODUCTION

Humans have the amazing capability of building useful abstractions that can capture regularities in the external world. Understanding what is responsible for this special feature of human intelligence relative to other animals is a longstanding goal in cognitive science (Penn et al., 2008; Berwick & Chomsky, 2016). One domain in which cognitive scientists have observed this "human singularity" (Dehaene et al., 2022) is in geometric reasoning: early *Homo sapiens* 100,000 years ago were able to produce structured abstract geometric shapes and drawings on caves (Henshilwood et al., 2011), whereas similar behaviors have not been observed for non-human primates despite years of human contact (Saito et al., 2014).

Such observations, as well as rigorous empirical work (e.g., Sablé-Meyer et al. 2021; 2022) have led some cognitive scientists to conclude that human mental representations uniquely contain discrete domain-specific symbols that are recursively and compositionally combined to produce abstractions that support the capacity for generalization that is characteristic of human behavior (Dehaene et al., 2022). A corollary of this hypothesis is that artificial neural networks cannot, in principle, produce human-like intelligence without the exogenous addition of explicit symbolic machinery and/or representations (Dehaene, 2021; Marcus, 2020). Indeed, empirical work in this domain has shown that explicitly symbolic models fit human behavior better than standard neural networks (Sablé-Meyer et al., 2021). This has led to the view, by some, that symbolic "Language of Thought" models are the best models of humans' mental representations (Quilty-Dunn et al., 2022).

However, the fact that human behavior, or their *inductive biases*, may be described effectively with abstract symbolic processing does not necessarily imply that their internal representations are based

on discrete symbols (Griffiths et al., 2023). Consequently, there may be other forms of representations, such as the continuous vector spaces of neural networks, that could, under the right conditions, produce this behavior without explicit symbolic machinery (McCoy et al., 2018). In the present work, we provide an existence proof of this point by revisiting recent empirical cognitive science work showing humans' regularity biases towards abstract geometric concepts (Sablé-Meyer et al., 2021; 2022). We show that standard neural networks augmented with a simple constraint that favors relational information processing can replicate human generalization and regularity biases without needing to build in explicit symbolic machinery. Specifically, we implement an architectural motif, known as the *relational bottleneck* (Webb et al., 2023a), that allows networks to exploit relations between objects rather than the attributes of individual objects.

We focus on the results of two studies. The first is the work of Sablé-Meyer et al. (2022), in which humans were tested on a standard working memory task, Delayed-Match to Sample (DMTS), using image stimuli sampled from a generative Language of Thought model of geometric concepts. The second is a study by Sablé-Meyer et al. (2021), in which humans and non-human primates were tested on a version of the Oddball Detection task, a simple categorization paradigm in which participants identify a deviant stimulus in a group of quadrilateral stimuli. We show that a standard neural network, augmented with a relational bottleneck and trained with an appropriately designed curriculum using the same data as the studies by Sablé-Meyer et al. (2021) and Sablé-Meyer et al. (2022), exhibited human-like biases for abstract geometric regularity. These results offer an alternative interpretation of such biases, suggesting that with the appropriate inductive biases and curriculum neural networks can exhibit features associated with the capacity for symbolic processing without the need to hardcode the network with symbolic representations and/or mechanisms.

## 2 HISTORICAL BACKGROUND AND RELATED WORK

For decades, cognitive scientists and AI researchers have embraced two main approaches to building intelligent systems: symbolic models (Fodor, 1975) and neural networks (Rumelhart & McClelland, 1986). Fodor (1975) proposed the "Language of Thought" (LoT) hypothesis: that higher-order cognition in humans is the product of recursive combinations of pre-existing, conceptual primitives, analogous to the way in which sentences in a language are constructed from simpler elements. Symbolic models are well-suited to naturally embed the abstract, structured knowledge humans possess, such as causal theories (Goodman et al., 2011) or hierarchical motor programs that draw handwritten characters (Lake et al., 2015). Neural networks, on the other hand, emphasize *emergence* of these abstract concepts purely from data within completely unstructured, distributed representations (McClelland et al., 2010). Despite the incredible recent success of neural networks in machine learning, cognitive scientists have hypothesized that their systematic failure at generalizing out of their training distribution comes from a failure to embed the kinds of abstract structural knowledge that can exist in symbolic models (Lake et al., 2017; Marcus, 2003).

Recent work has suggested that these capacities may emerge through learning in neural networks that implement *relational reasoning*. Relational reasoning involves abstracting over the details of particular stimuli or domains and extracting more general forms of structure that are broadly useful for capturing regularities in the external world (Gentner, 1983; Holyoak, 2012). This can be accomplished in neural networks by introducing an architectural inductive bias: the relational bottleneck (Webb et al., 2023a). The general principle of the relational bottleneck is that some components of the network are restricted to operating on relations over representations rather than the representations themselves (Webb et al., 2020; 2023b; Mondal et al., 2023). For example, the network might be constrained to use the similarity or distance between two embeddings rather than the embeddings themselves. Critically, unlike many hybrid neuro-symbolic models (Plate, 1995; Touretzky, 1990; Mao et al., 2019) the relational bottleneck does not introduce pre-specified symbolic primitives or any explicit mechanisms for symbolic processing, relying instead on the emergence of abstract concepts within unstructured, distributed representations. The motivation of the relational bottleneck is similar to that of other works that have built neural network architectures more sensitive to relational reasoning (Barrett et al., 2018; Santoro et al., 2017; Shanahan et al., 2020).

The Language of Thought (LoT) approach has been applied to a variety of domains in cognitive science, including learning causal theories (Goodman et al., 2011), representations of numbers (Piantadosi et al., 2012), and logical concepts (Piantadosi et al., 2016). However, geometry has recently

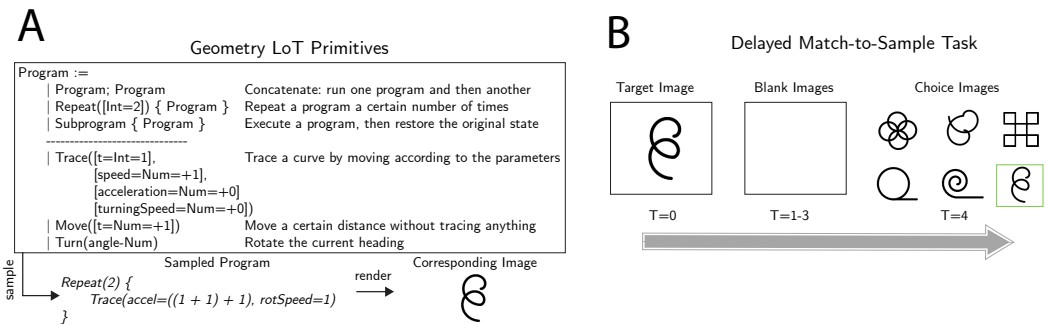

Figure 1: **Geometric Language of Thought and Delayed Match to Sample Task** (A) Primitives of the generative Language of Thought (LoT) model implemented in Sablé-Meyer et al. (2022). Primitives are recursively composed to produce symbolic programs that can be rendered into abstract geometric pattern stimuli. (B) Schematic of the working memory Delayed-Match to Sample (DMTS) task. A target stimulus is shown at the beginning, followed by a delay period, and the the target image must be selected out of a group of choice images containing distractors.

emerged as one of the domains in which the strongest arguments in favor of this kind of representation have been made (Sablé-Meyer et al., 2021; 2022; Dehaene et al., 2022). This setting is also a natural one in which to explore the predictions of neural network models, as geometric stimuli can be presented directly to models in the form of images. In the remainder of the paper, we present a detailed analysis of two of the studies that have been held up as providing support for the LoT approach, demonstrating how neural networks that are constrained to focus on relations are capable of reproducing the key patterns in human behavior.

## 3 TRAINING NEURAL NETWORKS ON A LANGUAGE OF THOUGHT FOR GEOMETRY

### 3.1 BACKGROUND

Sablé-Meyer et al. (2022) presented a study designed to test the Language of Thought hypothesis in the setting of geometry. The study was based on a model of geometric concept learning also developed by Sablé-Meyer et al. (2022). This model framed concept learning as program induction within the DreamCoder framework (Ellis et al., 2021). A base programming language was defined such that programs can be written to generate geometric shapes, where motor programs that draw geometric shapes are generated through recursive combination of symbolic primitives within a Domain Specific Language (DSL, Fig. 1A). The DSL contains motor primitives, such as tracing a particular curve and changing direction, as well as primitives to recursively combine subprograms such as $Concat$ (concatenate two subprograms together) and $Repeat$ (repeat a subprogram $n$ times). These symbolic programs can then be rendered into images such as the ones seen in Fig. 1. Since each image has an underlying program, the minimum description length (MDL; Ellis et al. 2021) of the program was used to model the psychological complexity of the corresponding geometric pattern.

Abstract geometric patterns were generated by this symbolic LoT model (Fig. 1A) and used as stimuli in a standard working memory task, based on a Delayed-Match to Sample (DMTS, Fig. 1B) paradigm. In this task, human participants were instructed to memorize a geometric stimulus. Following the memorization phase, participants were presented with a blank screen for two seconds. Subsequently, they were shown six option stimuli, among which one matched the original stimulus they had memorized (the target image), while the remaining five were distractors. The objective for participants was to accurately select the image they had seen during the encoding phase and avoid choosing any of the distractor images.

In preceding work (Sablé-Meyer et al. 2021, discussed further in the next section), the authors suggested that perception of abstract geometric stimuli can be based on two systems: a high-level,

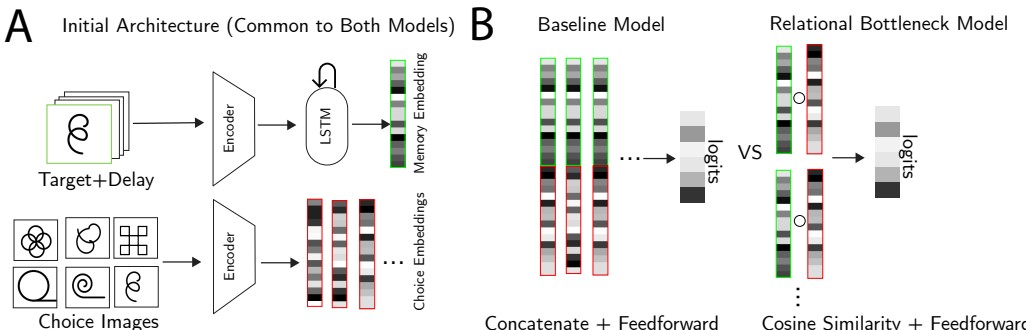

Figure 2: **DMTS Task Architecture Implementation** (A) Target and delay images are passed through a pretrained CNN encoder (Kubilius et al., 2019). The outputs of the encoder are passed to an LSTM, producing memory embeddings that correspond to participants' working memory representation of the initial target stimulus when performing the DMTS task. Each of the choice images are encoded using the same CNN encoder. (B) In the baseline model (left), the memory embeddings are simply concatenated to the choice embeddings and passed to a fully connected layer that produces the logits classifying the target image. In the relational bottleneck model (right), the embeddings are used to compute the similarity between each choice embedding and the memory embedding, and these similarities are used to produce the logits.

general-purpose symbolic system, supposedly only available to humans; and a lower-level, domain-specific shape invariant object recognition system, available to both humans and non-human primates, that can be modeled by a standard Convolutional Neural Network (CNN) model of object recognition in the brain (specifically, the Ventral Visual Stream; Kubilius et al. 2019). To study the first system, Sablé-Meyer et al. (2022) chose distractor stimuli that were maximally similar to the target image based on hidden representations of a pre-trained CNN model of the Ventral Visual system (CorNet; Kubilius et al. 2019) and the average grey-level of the image. Even with difficult distractors, humans excelled at the task, with error rates as low as $1.82\%$.

## 3.2 NEURAL NETWORK MODELING

We trained two Recurrent Neural Networks (RNNs; one baseline and one implementing a relational bottleneck) on this task, using the LoT model of (Sablé-Meyer et al., 2022) to generate a large training corpus of geometric stimuli and holding out the specific stimuli used in the human experiments for the test set. Stimuli were encoded by a CNN encoder, which was comprised of a pre-trained CNN model (CorNet; Kubilius et al. 2019). On each trial, an encoded representation of the stimulus was used as the input to an LSTM (Fig. 2A), followed by encoded representations of three additional timesteps-worth of blank input images[1] (Fig. 2A). The resulting output embedding of the LSTM corresponds to the working memory content of the human participants during choice time ("Memory Embedding", see Fig. 2A). The model is subsequently presented with the choice images (Fig. 2). We implemented two types of decision processes to classify the target image out of the six choice images (one target, five distractors). One of these was a standard baseline model, and the other was augmented with a relational bottleneck (Webb et al. 2023a; Fig. 2B).

For the baseline model, the embeddings of the six choice stimuli, along with the memory embedding, were concatenated and simultaneously fed into a standard feedforward layer that was used to classify the target image. For the Relational Bottleneck model, the cosine similarity between the memory embedding and each choice embedding was computed; those similarities were then used to produce the prediction of the target image. This restricted the model to processing the *relations* between its memory of the target image and the choice stimulus, without "intrusion" from any stimulus-specific attributes of the choice stimuli. During training, distractors were chosen randomly, but during testing, we used the exact same trials that were presented to human participants

---

[1]The delay period for the human experiments was 2 seconds, while the average stimulus presentation time was around 1.2s. Given this, we believe three timesteps makes the task for the networks at least as hard if not harder than the human task.

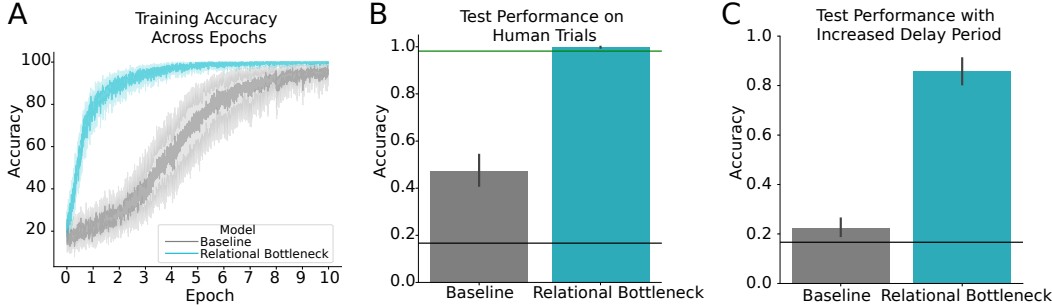

Figure 3: **DMTS Results** (A) Training accuracy across epochs of baseline and relational bottleneck models. Both models eventually reach near-perfect accuracy. (B) Results on tasks held out from model training that were taken directly from the human trials in Sablé-Meyer et al. (2022). The black bar denotes chance performance, while the green bar denotes mean human performance. Error bars are 95% confidence intervals over model training seeds. The Relational Bottleneck model performs much better out of distribution. (C) We increased the delay period from 3 timesteps to 20. Though both models suffer in performance, the Relational Bottleneck model still performs much better.

in the empirical study Sablé-Meyer et al. (2022), in which difficult distractors were chosen based on similarity to pretrained CorNet representations (Kubilius et al., 2019) and average grey-levels.

### 3.3 RESULTS

We tested both implementations of the model on the exact same trials given to human participants in Sablé-Meyer et al. (2022). Performance of the baseline model was well below human performance (Fig. 3B). However, the relational bottleneck model generalized extremely well to the test set, performing significantly better than the baseline model ($p < 0.001$) and approximating the performance of human participants. In addition, it handled longer delay periods substantially better than the baseline model (Fig. 3C), demonstrating its ability to maintain abstract representations of these geometric stimuli more robustly through the delay period. The results suggest that it is possible to achieve human-like performance on this task with a neural network model augmented by a simple constraint that favors learning relations, without imbuing the model with any explicit symbolic representations. The training corpus we used had stimuli containing very rich geometric abstractions (see Fig. 1A and Fig. 7). While our results suggest that inclusion of a relational bottleneck may be *necessary* to produce representations that support out-of-distribution generalization, it is not clear whether it is *sufficient* even in cases of a more impoverished training corpus.

Previous work has shown that a rich training data distribution can also contribute to such generalization (Chan et al., 2022). To address this, we tested whether the relational bottleneck would produce similar human-like performance when training on a relatively more restricted training corpus.

## 4 HUMAN-LIKE VS MONKEY-LIKE PROCESSING OF QUADRILATERAL STIMULI

### 4.1 BACKGROUND

Inspired by early anthropological work investigating abstract geometric concepts in cave drawings and behavioral research comparing geometric reasoning in humans and non-human primates, Sablé-Meyer et al. (2021) compared diverse human groups (varying in education, cultural background, and age) to non-human primates on a simple oddball discrimination task. Participants were shown a set of five reference shapes and one "oddball" shape and prompted to identify the oddball (Fig. 4). The reference shapes were generated based on basic geometric regularities: parallel lines, equal sides, equal angles, and right angles. Reference shapes consisted of 11 types of quadrilaterals varying in their geometric regularity, from squares (most regular) to random quadrilaterals containing no

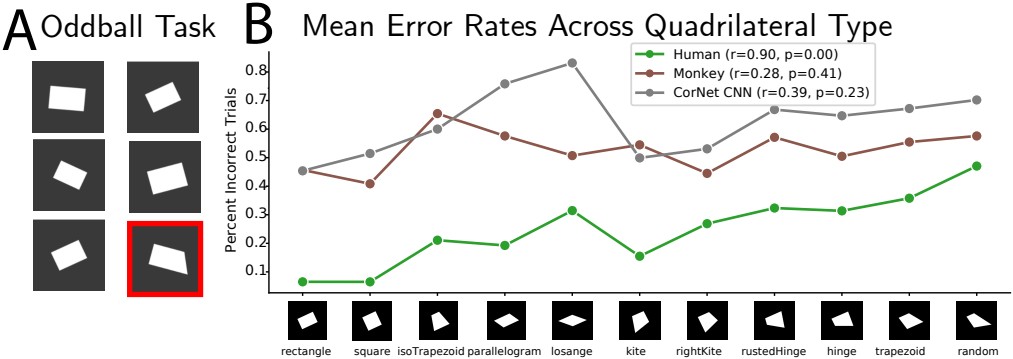

Figure 4: **Quadrilateral Oddball Task** (A) The Oddball task of Sablé-Meyer et al. (2021) used six quadrilateral stimulus images, in which five images were of the same reference shape (differing in scale and rotation) and one was an oddball (highlighted in red) that diverged from the reference shape's geometric properties. In this example, the reference shape is a rectangle; note that the Oddball does not have four right angles like the rectangles. (B) Sablé-Meyer et al. (2021) examined error rates for humans, monkeys, and pre-trained CNNs (Kubilius et al., 2019) across quadrilaterals of decreasing geometric regularity (from squares, which have the highest regularity, to random quadrilaterals that have little regularity). Humans performed significantly better on more regular images, with error rates trending significantly upwards with decreasing regularity, whereas monkey and CNN error rates did not exhibit a significant error rate trend as a function of regularity.

parallel lines, right angles, or equal angles/sides (least regular) (Fig. 4B). In each trial, five different versions of the same reference shape (e.g, a square) were shown in different sizes and orientations. The oddball shape was a modified version of the reference shape, in which the lower right vertex was moved such that it violated the regularity of the original reference shape (e.g, moving the lower right vertex of a trapezoid such that it no longer has parallel sides). Fig. 4A shows an example trial.

Sablé-Meyer et al. (2021) found that humans, across many different ages, cultures, and education levels, are naturally sensitive to these geometric regularities (right angles, parallelism, symmetry, etc) whereas non-human primates are not. Specifically, they found that human performance is best on the Oddball task for the most regular shapes, and systematically decreases as shapes become more irregular. Conversely, non-human primates perform well above chance, but they perform worse than humans overall and, critically, show no influence of geometric regularity (Fig. 4B).

To address this pattern of findings, Sablé-Meyer et al. (2021) implemented two computational models: a symbolic model and a neural network model. The symbolic model implemented oddball identification using an explicitly symbolic feature space constructed from the shapes' discrete geometric properties. The neural network model was a pretrained CNN model of the Ventral Visual stream (CORNet; Kubilius et al. 2019).[2] Sablé-Meyer et al. (2021) found that the symbolic model fit the human performance of their Oddball task significantly better than the neural network model, and in particular it captured the effect of increasing error with increasing geometric irregularity. Conversely, the neural network model fit the monkey behavior better, exhibiting no systematic relationship with the level of geometric regularity (Fig. 4B). They interpreted this as evidence that the human sensitivity to geometric regularity requires the presence of unique symbolic representations that are absent in both neural networks and non-human primates.

## 4.2 NEURAL NETWORK MODELING

Here, we show that a neural network trained on the same stimuli used by Sablé-Meyer et al. (2021), and provided with a relational bottleneck, exhibits the sensitivity of geometric regularity observed in humans, without the explicit specification of discrete symbolic representations.

---

[2]Sablé-Meyer et al. (2021) additionally re-trained CorNet on an object recognition task on the quadrilateral stimuli and reported that re-training CorNet on this task did not affect their results.

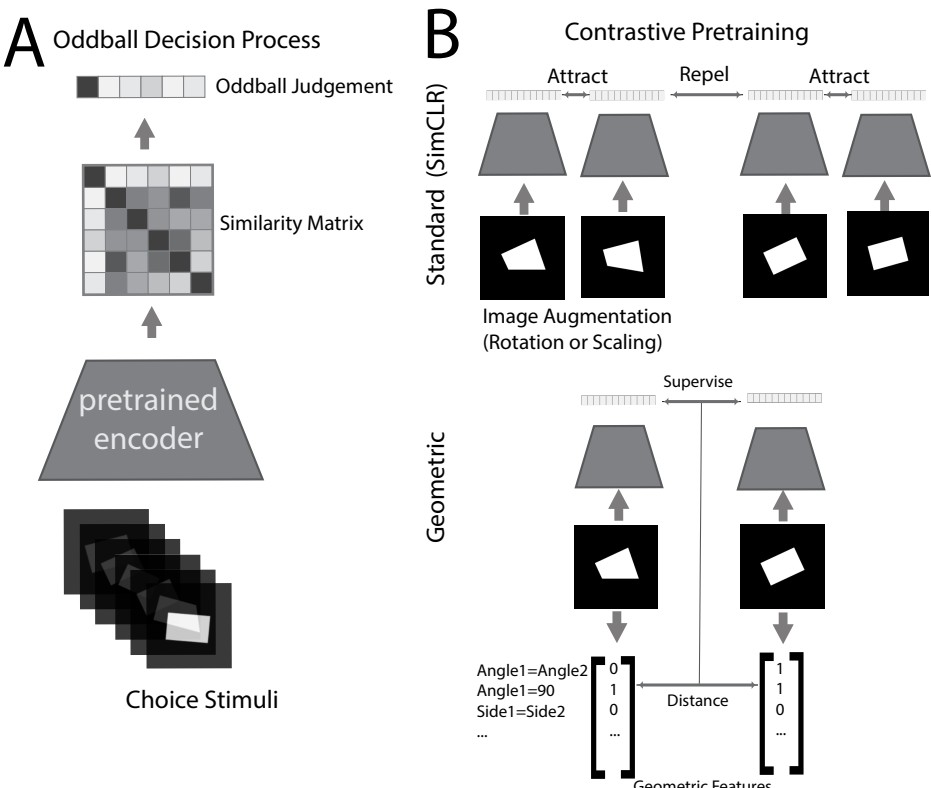

Figure 5: **Oddball Task Architecture Implementation** (A) To make an oddball decision using the Relational Bottleneck, we compute an oddball judgement directly from the $6 \times 6$ similarity matrix of the encoder's choice embeddings. (B) We implemented two types of contrastive pretraining on a ResNet CNN architecture: (top) a standard contrastive objective based on SimCLR (Chen et al., 2020) and (bottom) a novel contrastive objective using distances in a geometric feature space.

We started with the ResNet CNN architecture [3], but we modified this architecture to directly compute the Oddball judgements end-to-end using the relational bottleneck, using the method described in Kerg et al. (2022) (Fig. 5)A). Specifically, a $6 \times 6$ cosine similarity matrix is computed across each of the six stimuli, and the similarity matrix is fed into a feedforward layer that produces an Oddball decision. This structure forces the model to make decisions based on the relations between choice stimuli rather than the attributes of an individual choice stimulus.

We pretrained the CNN using one of two contrastive objectives (Fig. 5B): **Standard** and **Geometric**. The **Standard** objective was based on SimCLR (Chen et al., 2020). Specifically, simple random rotations and scaling were applied to individual quadrilateral images, and then the CNN was trained to push its representations of those images together, to be more similar (i.e., less distant) to their augmented counterparts, and pull its representations of different quadrilateral images apart, to be more dissimilar (i.e., more distant) from each other. The **Geometric** objective used the geometric features utilized in Sablé-Meyer et al. (2021) as the feature space over which to define distances. Those geometric features were binary vectors corresponding to the presence or absence of equal angles, equal sides, parallel lines, and right angles of the quadrilateral. During training, this effectively pushed quadrilaterals with similar geometric features together and pulled quadrilaterals with different geometric features apart. This allowed us to train the network to exhibit the same abstractions defined by the geometric features *without building in the geometric features themselves*. During testing and inference, the geometric features were completely discarded. This is similar to previous work instilling human biases into neural network agents (Kumar et al., 2022), in which the

---

[3]Note that although Sablé-Meyer et al. (2021) run their main experiments with CorNet, they show in their supplement that ResNet produces the same monkey-like behaviorial signatures as CorNet.

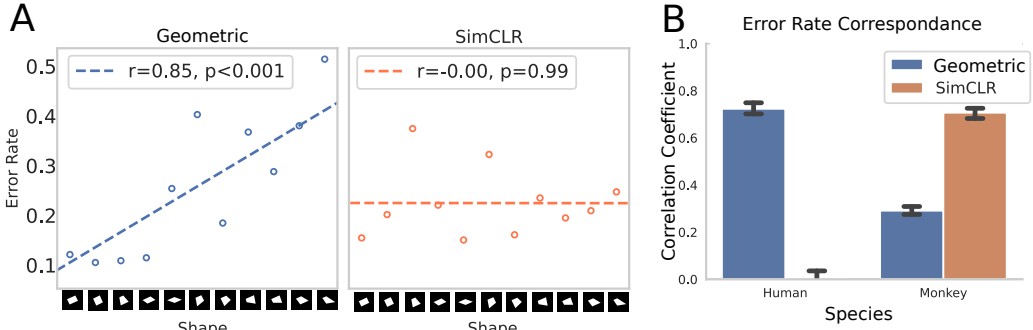

Figure 6: **Oddball Task Results** (A) Mean error rates over the 11 types of quadrilaterals for each type of network. The Geometric pre-trained network showed a significant trend between error rate and geometric regularity ($p < .001$), while the Standard (SimCLR) pre-trained network did not ($p = 0.99$). (B). We correlated error rates across quadrilaterals for each model with the corresponding error rates of humans and monkeys. Geometric pre-training of quadrilaterals led to human-like error patterns, whereas SimCLR pre-training led to more monkey-like error patterns. Error bars are 95% confidence intervals across different model training runs.

*tabula rasa* neural networks that were co-trained with symbolic information exhibited human biases without explicitly implementing any symbolic representations.

### 4.3 RESULTS

Similar to the effect observed in the study by Sablé-Meyer et al. (2022) discussed in the previous section, the geometric regularity effect observed for humans in Sablé-Meyer et al. (2021) was an inverse relationship between geometric regularity and error rate (see green plot in Fig. 4B). For example, humans performed best on the most regular shapes, such as squares and rectangles. This regularity effect was again absent in the monkey error rates (Fig. 4B).

Following Sablé-Meyer et al. (2021), we show, for each of our networks, the error rates for quadrilaterals sorted by geometric regularity and how well they match human and monkey error rates (Fig. 6). The Geometric pre-trained model showed a strong fit to human behavior ($r = 0.72$) and a significant effect of geometric regularity ($p < 0.001$; Fig. 6). The Standard (SimCLR) pre-trained model, however, showed a strong fit to *monkey* behavior ($r = 0.70$), but not to *human* behavior ($r = 0.005$), nor did they show the geometric regularity effect ($p = 0.99$; Fig. 6). This indicates that, although the relational bottleneck was necessary, it was not sufficient on its own to reproduce human behavior on this task. However, coupled with the appropriate training, it was able to reproduce the pattern of results observed for human behavior in Sablé-Meyer et al. (2021). These results suggest that, with the appropriate structural biases and training experience, it is possible for neural network to learn representations that exhibit human-like biases in the geometric oddball task without explicitly imposing symbolic representations on the network.

## 5 DISCUSSION

A prevailing theory in cognitive science is that abstractions that support strong generalization reflect the presence of symbolic systems innate in humans that may be absent in animals (Fodor, 1975; Quilty-Dunn et al., 2022; Dehaene et al., 2022). Along similar lines, it has been argued that, without explicitly imbuing neural networks with such capabilities, they will not be able to exhibit the same cognitive flexibility as humans (Marcus, 2020; Dehaene, 2021). Empirical findings in the studies by (Sablé-Meyer et al., 2021) and Sablé-Meyer et al. (2022) have been offered in support of these conjectures. Here, we provide evidence to the contrary, showing how the introduction of a simple, neurally plausible relational inductive bias, coupled with the appropriate training experiences, is

sufficient to reproduce behavior consistent with the formation of abstract representations in neural networks.

The domain of the empirical work we re-examine involves the visual perception of geometric patterns (Sablé-Meyer et al., 2021; 2022). Sablé-Meyer et al. (2022) show that humans are adept at processing geometric patterns, using a delayed-match-to-sample working memory task with stimuli sampled from a generative probabilistic program induction model (Ellis et al., 2021). We trained two types of RNN models on this task: a baseline model and a model with a relational bottleneck that is biased to focus on *relations between stimuli* to classify the target image. Consistent with the claims of Sablé-Meyer et al. (2022), a baseline model does not reach human-level performance out of its training distribution. However, a model with the relational bottleneck does indeed reach human performance on the test set, showing that a simple constraint that favors learning relations can allow neural networks to achieve human-level performance on this task.

Sablé-Meyer et al. (2021) further show that humans are sensitive to geometric regularity when performing a visual perception task, the Oddball task, using quadrilateral stimuli, whereas non-human primates and standard CNNs (Kubilius et al., 2019) are not. Here, we found that even with a relational bottleneck, a network trained with a standard contrastive learning objective produced the same monkey-like behavior observed from the CNN trained by Sablé-Meyer et al. (2021). However, when trained contrastively on distances produced by geometric features, the model did reproduce the human geometric regularity effect.

One important difference between the two tasks is that, the delayed match to sample task Sablé-Meyer et al. (2022) used reaction times (RTs) to show the geometric regularity effect in humans, whereas the oddball task Sablé-Meyer et al. (2021) used error rates. This is because error rates in the former were near zero, and therefore RTs were required to observe significant effects. One limitation of our study is that we did not construct an analogue to human RTs for our RNN models. Instead, we used out-of-training-distribution accuracy as the main performance metric. In the Oddball task (Sablé-Meyer et al., 2021), where human error rates were higher, we were able to conduct a more direct comparison, where we observed a clear correspondence between human (or monkey) behavior and our models.

A further difference between the two experiments is that the model of the Oddball task required geometric contrastive pre-training to match human performance (producing monkey-like behavior without this objective). We believe this is because the dataset used in the Delayed Match-to-Sample task features a richer distribution of stimuli (Fig. 7) sampled from a Bayesian program induction model (DreamCoder; Ellis et al. 2021). Building a training distribution of samples from such a Bayesian model has an interpretation of effectively distilling the Bayesian model's rich prior into a neural network (McCoy & Griffiths, 2023). In contrast, the Oddball dataset consisted of a relatively simple set of 11 quadrilaterals, which may not be sufficiently diverse to allow the network to extract more abstract representations (see Chan et al. 2022 for a similar argument about how the richness of training data affects the post-training capabilities of Large Language Models).

Our work provides evidence that simple modifications to standard neural networks are sufficient to reproduce human behavior on tasks used in cognitive science to showcase allegedly unique human capabilities. It may be possible that such geometric regularity biases can be instilled in neural networks by other methods. For example, previous work has shown Vision Transformer architectures, like humans, are biased more towards shapes than textures (Tuli et al., 2021). In general, we suggest that human-like behavior and abstractions can be instilled in neural networks using a variety of strategies, including through specialized architectures (Webb et al., 2023a; 2020), specialized loss functions/training curricula (Kumar et al., 2022; Kepple et al., 2022), and/or highly rich data distributions (McCoy & Griffiths, 2023; Chan et al., 2022).

A hallmark of human intelligence is the ability to develop highly general abstractions that capture the essential structure in their environments in a strikingly sample-efficient manner (Gershman, 2017; Lake et al., 2017). Our work highlights the possibility of neural network-based architectures achieving the same level of intelligence without built-in, explicitly symbolic machinery, recapitulating a classic debate in cognitive science (Rumelhart & McClelland, 1986). Given the success of this approach in the geometric setting, we anticipate that similar models may be able to capture behavior that has previously been explained in terms of symbolic representations in learning causal relationships, numerical representations, and logical concepts.

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

# A APPENDIX

## A.1 MORE GEOMETRIC LOT STIMULI

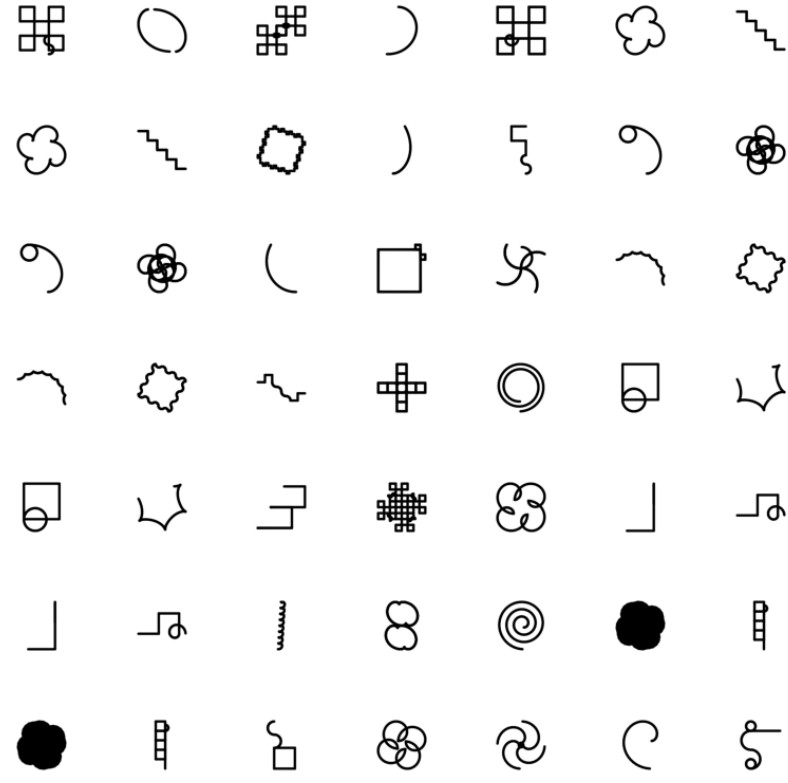

Figure 7: More examples of samples from the LoT model in Sablé-Meyer et al. (2022) (see Fig. 1)

A.2 ODDBALL HUMAN/MONKEY CORRESPONDANCE OVER TRAINING EPOCHS

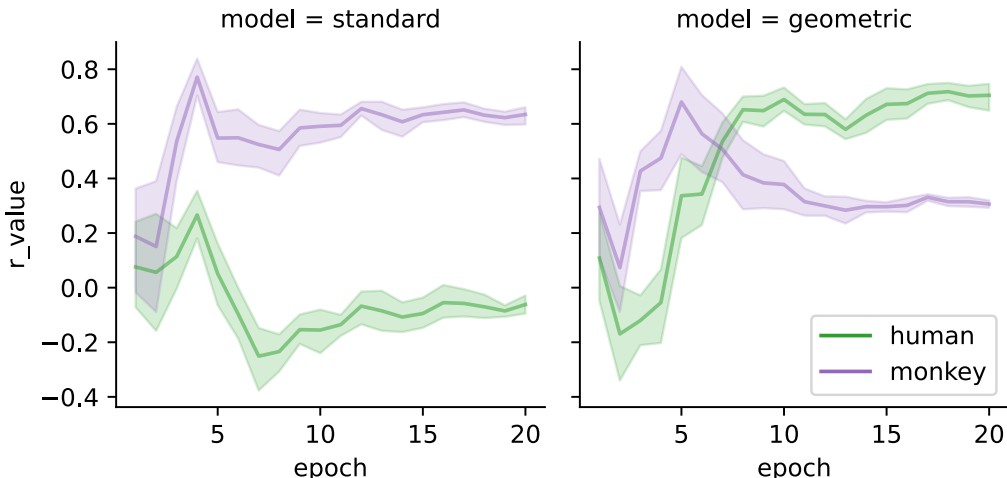

Figure 8: Correlation between Oddball model error rates and human/monkey error rates across contrastive training epochs. During geometric contrastive pre-training (see Fig. 5), there is an inflection point in which the model becomes more human-like and less monkey-like.

