# OpenReview forum: "Relational Constraints On Neural Networks Reproduce Human Biases towards Abstract Geometric Regularity"
_ICLR.cc/2024/Conference — ICLR 2024 Conference Withdrawn Submission_

### Official Review · Reviewer_W6kC · 2023-10-31

**Soundness:** 2 fair
**Presentation:** 3 good
**Contribution:** 2 fair
**Rating:** 5
**Confidence:** 3

**Summary:**

The paper challenges the prevailing symbolic ``Language of Thought'' (LoT) hypothesis for neural networks to possess human-like bias towards geometric regularity. It's believed that humans show bias towards regular and symmetric forms compared to geometrically irregular counterparts because of the existence of discrete symbolic structure in human mental representation, and the similar conclusion shall apply to neural networks. In this work, the authors counter-argue the hypothesis by showing the with the relational bottleneck and proper training scheme, neural networks without explicit symbolic structure show similar bias like humans in tasks of delayed match to sample and oddball.

**Strengths:**

Originality: To the best of my knowledge, this is an original work, it argues against a common belief among the community by showing that without the structure, a model could still achieve human-like performance.

Quality: The paper is of good quality with good experimental evaluation and careful setups to prove its argument.

Clarity: The paper is clearly presented with sufficient background, model details and experimental results.

Significance: The work may be of limited significance as the work doesn't really contribute practical advances now that the community has shown extremely good performance among a variety of tasks when a purely neural model is scaled without symbolic structures.

**Weaknesses:**

1. On significance, as I mentioned above, I doubt whether the existence of symbolic structures would be of significant interest to the community, or the issue on symbolic structures needs debating, now that gigantic models like GPT-4 without any symbolic structures have already managed to achieved quite well on a variety of tasks, conventionally believed to require symbolic understanding.
2. As the model requires generated data, I'm concerned that the training set may be contaminated. How can you make sure that your test data is never in your training set? Or extremely similar cases are in your training set? If you cannot make sure of that, then your results might not be reliable, as the model showing better performance might simply be better at memorizing rather than generalizing.
3. Regarding the concatenate and feedforward vs cosine and feedforward, have you added non-linearity between feedforwards? A linear method cannot approximate cosine, but a MLP could theoretically, and your model design simply bypasses that. Could it be possible that simple models without relational bottleneck could be as good as those with relational bottleneck, but just slower in learning?
4. Both relational bottleneck and contrastive and similar conclusions have been verified in Webb et al.'s works. So in this perspective, how is this specific work different from the general conclusions drawn from them?

**Questions:**

See above.

---

### Official Review · Reviewer_F4KU · 2023-10-31

**Soundness:** 3 good
**Presentation:** 2 fair
**Contribution:** 1 poor
**Rating:** 3
**Confidence:** 3

**Summary:**

In this manuscript the authors present biases that make neural networks reproduce two previous results that were interpreted as evidence for symbolic representations or language of thought in the original papers. Essentially, this is a refutation of the argument presented in the earlier papers by showing that neural networks without a symbolic representation can still solve the tasks in a human-like way.

**Strengths:**

I do not disagree with the content of this manuscript much and think that most of the machine learning community would have expected that the results observed here could be reproduced by a deep neural network with the right biases. Also the neural networks appear to be relatively clear counterexamples to the argument about the necessity of a language of thought.

**Weaknesses:**

This paper appears somewhat misplaced at ICLR. It seems much more sensible to continue the conversation in the journals where the original studies and claims about language of thought were published than to present this to the ICLR community where differences between monkey and human representations about symmetry are an extremely niche or absent topic and the language of thought is certainly not a prevailing hypothesis. Additionally, a response to specific papers should be published such that it is read by a similar audience and is naturally of fairly narrow interest.

In terms of the content, I personally do not view these experiments as particularly strong evidence for either form of representation. Once you go to complex representations that can interact, I don’t think there is a clean separation between deep features and approximations to a formal language of thought. Ultimately, we expect this language to be encoded in neural responses. For the evidence in this model a critical evaluation what aspects of the language of thought model and of the deep neural networks are truly necessary would be good I think.
None of this seems relevant for the ICLR community though.

**Questions:**

My main questions would be: Why ICLR?

---

### Official Review · Reviewer_eSYk · 2023-11-01

**Soundness:** 2 fair
**Presentation:** 3 good
**Contribution:** 3 good
**Rating:** 5
**Confidence:** 5

**Summary:**

The authors are tackling the question of whether tasks usually considered within the purview of symbolic models can be performed by artificial neural networks. They specifically target geometric intuition in humans and non-human primates. They show that certain models with a "relational bottleneck" can replicate performance patterns on geometric tasks.

**Strengths:**

The work tackles a question of popular interest (symbolic vs connectionist models) and is well-motivated

It explores the utility of a relatively simple and easy to understand model component (relational bottleneck)

The paper is clearly written and the analyses are clear

**Weaknesses:**

I feel that the experiments performed don't necessarily allow for strong conclusions about what is necessary/sufficient for performing these tasks. With respect to the DMTS experiment, the authors say "inclusion of a relational bottleneck may be
necessary to produce representations that support out-of-distribution generalization". But with comparison to only one very simple alternative model the evidence for this is very weak. Was there any hyperparameter optimization done for the baseline model? Were other architectures explored, e.g. what if several MLP layers were allowed between the encoding and the logits? More problematic, with the oddball task, there is no comparison to similar networks without the relational bottleneck. The authors should test the same pre-trained encoders within architectures without a RB to see if the same trend found with the DMTS is replicated here.

It is unclear to me whether the relational bottleneck is meant to be important because of the computation it does (cosine sim) or because of the representations that are learned in a network that has it. In these tasks, where a measure of similarity is the end goal, it seems like the RB could be useful even if it was just added posthoc as a means of comparing representations of the stimuli, even without any further training. It would be good if the authors were clear on what the intended reason is for the RB being useful and possible do experiments to validate it (e.g. by adding a cosine sim after a pretrained encoder and using the resulting similarity values to identify oddballs directly rather than adding a supervised layer).

With humans near ceiling performance on the DMTS task, there isn't much signal to determine if the model behaves in a human-like way (beyond just performing well). It is mentioned that the original DMTS data includes reaction time effects, and these could be studied in the model using logit values as a proxy, which could provide stronger evidence to determine which models are a better fit.

**Questions:**

Why is the symbolic model's performance not also on fig 4? It would be useful context.

---

### Official Review · Reviewer_UjxR · 2023-11-06

**Soundness:** 2 fair
**Presentation:** 3 good
**Contribution:** 2 fair
**Rating:** 5
**Confidence:** 3

**Summary:**

This paper investigates two alternative hypotheses for the computational architectures underlying human behavior in 2 visual perception tasks. For instance, psychophysics experiments have shown that human visual perception is biased by the regularity of geometric shapes such that some types of recognition tasks are easier for regular shapes than for irregular ones. Some previous work has argued that this sensitivity to regularity implies that this type of tasks are being solved by humans through mental representations of discrete symbolic structure, concluding that the brain will have to implement some form of symbolic processing. This paper argues on the other hand that there is a way to naturally recover the recorded types of visual perception bias within a connectionist neural network model through an architectural tweak that implements a "relational structure": essentially after the visual inputs are feturized through a pretrained vision backbone, they are passed through a module that computes the pairwise cosine similarity between them and all the downstream processing is then performed on the resulting pairwise similiarity matrix. The paper shows that this architecture embedded in a neural network model recovers the sensitivity to geometric regularity displayed by human subject and previously imputed to a symbolic processing, providing an existence proof that disputes the prevailing symbolic model of geometric reasoning.

**Strengths:**

- The paper is well written and clear to follow
- The relational bottleneck model is interesting and potentially useful as a general module in neural network architectures

**Weaknesses:**

- The paper makes some specific idiosyncratic choices in terms of architecture (LSTM vs GRU or plain RNN, concatenations instead of some type of projection, number of layers, choice of pretrained backbones, self-supervised learning algorithm, etc) which are hardly motivated in a principled way
- The relational representation strongly depends on the vision backbone and the geometry over the featurized patterns that they induce, but this is not analyzed and could at least be discussed
- There seems to be a big discrepancy in terms of number of parameters between the relational bottleneck model and the baseline neural network models. In particular, for instance the outputs in the first use case are generated by operating on a 6 x 6 similarity matrix vs the full-dimensional concatenated activation patterns
- In relation to the previous point, the paper does not discuss and analyses the effects of chances in capacity and sample complexity due to the relational bottleneck. There's an argument to be made that some of the observed results might be due at least in part to a statistical effect due to the different inductive biases and regularization coming from the bottleneck
- The whole debate putting symbolic processing and connectionist models in opposition in terms of explaining brain processes seems a bit contrived. On one hand, the brain is obviously a connectionist architecture, its main computational substrate being neurons and synapses. On the other hand, RNNs are Turing complete, and therefore can emulate any symbolic algorithm (given enough resources). Moreover, there's hardly any opposition to the notion that a "symbolic" apparatus like a digital computer can be used to emulate a connectionist model like a neural network. From these perspectives, the debate on symbolic vs neural network architectures seems arguably artificial and sterile. It would be beneficial in order to motivate the contribution of this paper if the authors could push back on this (possibly naive and superficial) observations and make the case that addressing this debate would be benefiting the broad ICLR community

**Questions:**

- How robust are the main results over the choice of vision backbones?
- Would it be possible to (at least approximately) match the number of free parameters of the different models in order to address the concerns above?
- The idea of a relational bottleneck seems to be related to the literature on disentangled and parallel representations? If that makes sense, it would be potentially interesting to have a brief references to these alternative methods to incorporate "semantic properties" in terms of the geometry of the activation patterns